# 3D Copy-Paste: Physically Plausible Object Insertion for Monocular 3D Detection

**Yunhao Ge**◇†, **Hong-Xing Yu**◇, **Cheng Zhao**§, **Yuliang Guo**§, **Xinyu Huang**§, **Liu Ren**§,
**Laurent Itti**†, **Jiajun Wu**◇

◇Stanford University    †University of Southern California
§Bosch Research North America, Bosch Center for Artificial Intelligence (BCAI)
{yunhaoge, koven, jiajunwu}@cs.stanford.edu  {yunhaoge, itti}@usc.edu
{Cheng.Zhao, Yuliang.Guo2, Xinyu.Huang, Liu.Ren}@us.bosch.com

## Abstract

A major challenge in monocular 3D object detection is the limited diversity and quantity of objects in real datasets. While augmenting real scenes with virtual objects holds promise to improve both the diversity and quantity of the objects, it remains elusive due to the lack of an effective 3D object insertion method in complex real captured scenes. In this work, we study augmenting complex real indoor scenes with virtual objects for monocular 3D object detection. The main challenge is to automatically identify plausible physical properties for virtual assets (e.g., locations, appearances, sizes, etc.) in cluttered real scenes. To address this challenge, we propose a physically plausible indoor 3D object insertion approach to automatically *copy* virtual objects and *paste* them into real scenes. The resulting objects in scenes have 3D bounding boxes with plausible physical locations and appearances. In particular, our method first identifies physically feasible locations and poses for the inserted objects to prevent collisions with the existing room layout. Subsequently, it estimates spatially-varying illumination for the insertion location, enabling the immersive blending of the virtual objects into the original scene with plausible appearances and cast shadows. We show that our augmentation method significantly improves existing monocular 3D object models and achieves state-of-the-art performance. For the first time, we demonstrate that a physically plausible 3D object insertion, serving as a generative data augmentation technique, can lead to significant improvements for discriminative downstream tasks such as monocular 3D object detection. Project website: https://gyhandy.github.io/3D-Copy-Paste/.

## 1 Introduction

Monocular indoor 3D object detection methods have shown promising results in various applications such as robotics and augmented reality [Yang and Scherer, 2019, Chen et al., 2017]. However, the deployment of these methods is potentially constrained by the limited diversity and quantity of objects in existing real datasets. For example, in SUN RGB-D dataset [Song et al., 2015], the bathtub category has only less than 500 annotations compared to chair which has over 19,000 annotations. This may be due to the difficulty in acquiring and labeling substantial indoor scene datasets with diverse 3D object annotations [Silberman et al., 2012, Song et al., 2015, Dai et al., 2017].

Data augmentation techniques have been widely utilized in 2D detection and segmentation tasks to improve the diversity and quantity of the available training data [Dwibedi et al., 2017, Ge et al., 2022a, Ghiasi et al., 2021, Ge et al., 2022b, 2023]. However, it is non-trivial to scale 2D augmentation methods to 3D scenes due to physical constraints in real 3D scenes. In particular, technical challenges

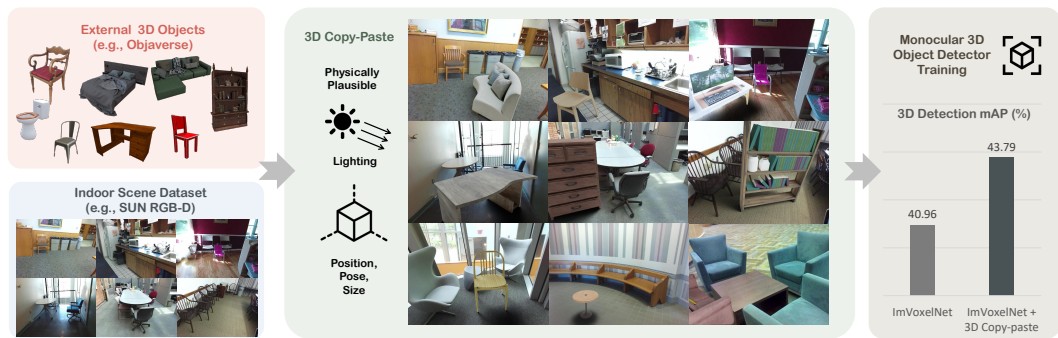

Figure 1: Overall pipeline of physically plausible object insertion for monocular 3D object detection: Our approach *copies* external 3D objects (e.g., from Objaverse [Deitke et al., 2022]) and *pastes* them into indoor scene datasets (e.g., SUN RGB-D [Song et al., 2015]) in a physically plausible manner. The augmented indoor scene dataset, enriched with inserted 3D objects, is then used to train monocular 3D object detection models, resulting in significant performance improvements.

emerge especially in how to maintain physical plausibility for: (1) **Collision and Occlusion Handling**: In 3D data augmentation, handling collisions between objects is more challenging than in 2D data. Properly managing collisions is essential to prevent artifacts and ensure that objects appear as natural and coherent parts of the scene. (2) **Illumination and Shading**: For 3D data, augmenting objects requires careful consideration of the lighting conditions in the scene to create realistic shading and reflections. This involves estimating the spatially-varying illumination and adapting the appearance of the inserted objects to maintain visual coherence. (3) **Geometric Consistency**: In 3D data augmentation, maintaining geometric consistency is crucial to ensure that the augmented objects fit naturally within the scene. Unlike 2D augmentation, which deals with flat images, 3D augmentation must consider spatial relationships, object orientations, and their interaction with the surrounding environment.

In this paper, we explore a novel approach, 3D Copy-Paste, to achieve 3D data augmentation in indoor scenes. We employ physically plausible indoor 3D object insertion to automatically generate large-scale annotated 3D objects with both plausible physical location and illumination. Unlike outdoor scenarios, indoor environments present unique challenges: (1) complex spatial layouts, notably cluttered backgrounds and limited space for object placement, which require a meticulously crafted method for automated object positioning (ensuring realistic position, size, and pose), and (2) intricate lighting effects, such as soft shadows, inter-reflections, and long-range light source dependency, which necessitate sophisticated lighting considerations for harmonious object insertion.

Fig. 1 shows our overall pipeline. In our approach, we take advantage of existing large-scale 3D object datasets, from which we *copy* simulated 3D objects and *paste* them into real scenes. To address the challenges associated with creating physically plausible insertions, we employ a three-step process. First, we analyze the scene by identifying all suitable planes for 3D object insertion. Next, we estimate the object's pose and size, taking into account the insertion site to prevent collisions. Lastly, we estimate the spatially-varying illumination to render realistic shading and shadows for the inserted object, ensuring that it is seamlessly blended into the scene.

Our proposed method augment existing indoor scene datasets, such as SUN RGB-D [Song et al., 2015], by incorporating large-scale 3D object datasets like Objaverse [Deitke et al., 2022] using our 3D Copy-Paste approach. Our method is an offline augmentation method that creates a new augmented dataset. The monocular 3D object detection model, ImvoxelNet Rukhovich et al. [2022], trained on this augmented dataset, achieves new state-of-the-art performance on the challenging SUN RGB-D dataset. We systematically evaluate the influence of the inserted objects' physical position and illumination on the downstream performance of the final monocular 3D object detection model. Our results suggest that physically plausible 3D object insertion can serve as an effective generative data augmentation technique, leading to state-of-the-art performances in discriminative downstream tasks such as monocular 3D object detection.

We make three main contributions: (1) We introduce 3D Copy-Paste, a novel physically plausible indoor object insertion technique for automatically generating large-scale annotated 3D objects. This

approach ensures the plausibility of the objects' physical location, size, pose, and illumination within the scene. (2) We demonstrate that training a monocular 3D object detection model on a dataset augmented using our 3D Copy-Paste technique results in state-of-the-art performance. Our results show that a physically plausible 3D object insertion method can serve as an effective generative data augmentation technique, leading to significant improvements in discriminative downstream monocular 3D object detection tasks. (3) We conduct a systematic evaluation on the effect of location and illumination of the inserted objects on the performance of the downstream monocular 3D object detection model. This analysis provides valuable insights into the role of these factors in the overall effectiveness of our proposed approach.

## 2 Related Works

### 2.1 Monocular 3D Object Detection

Monocular 3D Object Detection estimates the 3D location, orientation, and dimensions (3D bounding box) of objects from a single 2D image. It has garnered significant attention in recent years due to its potential applications in autonomous driving, robotics, and augmented reality. There are many works of monocular 3D detection in driving scenarios, such as 3DOP[Chen et al., 2015], MLFusion[Xu and Chen, 2018], M3D-RPN[Brazil and Liu, 2019], MonoDIS[Simonelli et al., 2019], Pseudo-LiDAR[Wang et al., 2019], FCOS3D[Wang et al., 2021], SMOKE[Liu et al., 2020], RTM3D[Li et al., 2020a], PGD[Wang et al., 2022a], CaDDN[Reading et al., 2021]. Specifically, Geometry-based Approaches: MV3D Chen et al. [2017] utilized both LiDAR-based point clouds and geometric cues from images for 3D object detection. Mousavian et al. [2017] introduced a method that regresses object properties such as dimensions, orientation, and location from 2D bounding boxes using geometric constraints. In the context of indoor scenes, multi-task learning has gained traction. Recent studies, including PointFusion by Xu et al. [2018], have amalgamated 3D object detection with tasks like depth estimation or semantic segmentation to improve performance. Total3D [Nie et al., 2020] and Implicit3D [Zhang et al., 2021] use end-to-end solutions to jointly reconstruct room layout, object bounding boxes and meshes from a single image. ImvoxelNet [Rukhovich et al., 2022] achieves state-of-the-art performance by using the image-voxels projection for monocular 3d object detection.

### 2.2 3D Data Augmentation

Data augmentation in 3D has become increasingly vital for enhancing performance across various 3D perception tasks. Most of work focuses on outdoor scenes [Zhang et al., 2020, Lian et al., 2022, Abu Alhaija et al., 2018, Chen et al., 2021, Tong et al., 2023]. Geometric Transformations: Wu et al. [2015] applied rotations, translations, and scaling to augment the ModelNet dataset, improving classification and retrieval tasks. Point Cloud Augmentation: Engelcke et al. [2017] proposed techniques such as random point removal, Gaussian noise, and point cloud interpolation for augmenting LiDAR datasets, enhancing object detection and segmentation performance. Generative Model-based Augmentation: Smith and Meger [2017] used a conditional GAN to generate diverse and realistic 3D objects. Similarly, Achlioptas et al. [2018] employed a VAE for learning a generative model of 3D shapes for shape completion and exploration tasks. However, while 3D generative models can achieve object-level augmentation, they are not scalable to scene-level augmentation. 2D generative models can produce highly realistic images, but they do not provide physically plausible 3D labels. 3D Common corruptions [Kar et al., 2022] use 3D information to generate real-world corruptions for 2D dataset, which can evaluate the model robustness and be used as a data augmentation for model training, but does not support 3D detection because it does not introduce new 3D object content.

### 2.3 Illumination Estimation

Illumination estimation is a critical focus within computer vision research, given its crucial role in various applications. Li et al. [2020b] addressed the inverse rendering problem for complex indoor scenes, estimating spatially-varying lighting, SVBRDF, and shape from a single image. Meanwhile, a differentiable ray tracing method combined with deep learning was proposed for the learning-based inverse rendering of indoor scenes [Zhu et al., 2022]. Additionally, research has been conducted on using deep learning for indoor lighting estimation, with methods like Deep Parametric Indoor Lighting Estimation offering enhanced accuracy and efficiency Gardner et al. [2019]. Furthermore,

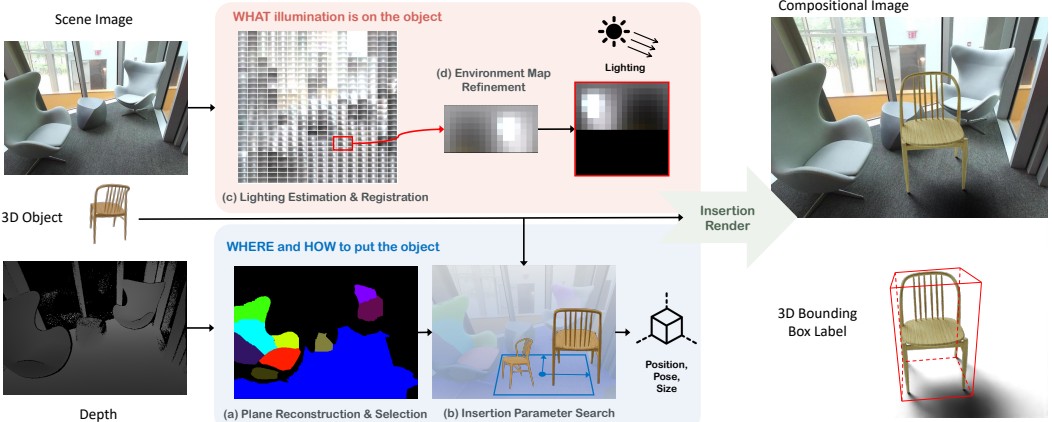

Figure 2: 3D Copy-Paste method overview: Our method (a) processes the input RGB image and depth data to reconstruct floor planes that can accommodate inserted objects. (b) Using the reconstructed planes and information about objects in the original scene, we estimate a physically plausible position, pose, and size for the inserted objects, ensuring they do not collide with existing objects. (c) We predict the spatially-varying lighting of the scene. (d) By registering the insertion position determined in (b) to spatially-varying lighting, our light estimation module (d) refined an HDR environment map to represent the lighting information for the inserted objects. (e) The insertion rendering module takes the position, pose, size, and lighting as input and inserts a 3D object into the real scene, adjusting the object's lighting and shadows accordingly to ensure it seamlessly integrates as a natural and coherent part of the scene.

Wang et al. [2022b] introduced Neural Light Field Estimation, a method that effectively models complex lighting conditions for virtual object insertion in street scenes. These studies underscore the potential of machine learning in improving illumination estimation capabilities in rendering and computer vision tasks.

# 3   Methods

This section presents our proposed physically plausible indoor 3D object insertion approach. Fig. 2 shows our 3D Copy-Paste method overview. Section 3.1 addresses the question of "where and how to place the object", detailing the process of estimating suitable insertion positions, poses, and sizes for the objects while avoiding collisions with existing objects. Section 3.2 explains "what illumination should we add to the object": estimate the scene's spatially-varying illumination and render the inserted objects with realistic lighting and shadows. Section 3.3 describes how we create an augmented dataset using the inserted objects and train monocular 3D object detection models.

## 3.1   *Where and how*: Physically Plausible Position, Pose, and Size Estimation

This section describes handling the first challenge of avoiding collisions during insertion by estimating physically plausible position, pose, and size parameters.

### 3.1.1   Ground Plane Selection

Given a scene and a 3D object to insert, the initial question is where to place the object. To accommodate a new object, we must identify and understand the available regions where the object can be situated. We perform plane reconstruction to comprehend the scene's layout and subsequently, we estimate physically plausible key parameters such as position, size, and pose. Fig. 2(a) presents an overview of our plane reconstruction and selection module, which takes an RGB image and depth data as input and predicts all potential planes, then narrows down to the ground plane.

To get a rough plane reconstruction, we followed the plane extraction method using Agglomerative Hierarchical Clustering (AHC) described in Feng et al. [2014]. There are three main steps: (1)

we construct a graph with nodes and edges representing groups of points, obtained by dividing the point cloud (merging RGB with depth) into non-overlapping groups. (2) We then perform AHC on the organized graph to identify potential planes by merging nodes that belong to the same plane, continuing until the mean squared error of plane fitting surpasses a threshold. (3) We use a pixel-wise region-growing method to refine the detected planes. To further refine the extracted planes while preserving clear face textures and sharp features without losing geometric details, we utilize a back-end indoor plane optimization and reconstruction method described in Wang and Guo [2018]. Specifically, we first partition the entire dense mesh into different planar clusters based on the planes extracted with AHC, treating them as plane primitives. We then create a texture patch for each plane and sample points on it, followed by executing a global optimization process to maximize the photometric consistency of sampled points across frames by optimizing camera poses, plane parameters, and texture colors. Further, we optimize the mesh geometry by maximizing consistency between geometry and plane primitives, further preserving the original scene's sharp features, such as edges and corners of plane intersections. Finally, we get the reconstructed plane with the geometry parameters (e.g., surface normal).

To select a proper plane for insertion, we first identify all horizontal planes based on surface direction and the standard deviation along the Z-axis. Specifically, there are two constraints for considering a plane as horizontal: (1) The plane must have a surface normal aligned with the positive direction of the Z-axis (opposite of the gravity vector), and (2) the standard deviation along the Z-axis should be smaller than a predefined threshold. In our scenario, we aim to insert furniture into the scene, such as the ten interest classes in the SUN RGB-D dataset [Song et al., 2015]: sofa, bed, chair, desk, table, nightstand, dresser, bookshelf, toilet, and bathtub. Consequently, we must identify the floor plane by selecting the horizontal plane with the lowest average Z value among all detected horizontal planes.

### 3.1.2 Constrained Insertion Parameter Search

To address the question of where and how to place the object, we estimate specific insertion parameters: position ($p$), size ($s$), and pose ($o$). We propose an efficient constrained insertion parameter searching algorithm to calculate plausible insertion parameters while avoiding collisions with existing objects in the scene (Algorithm 1). Given the reconstructed floor plane, we first determine the search space for each parameter. For position, we want the inserted object to touch the floor, so we find the 3D bounding box of the object and calculate the center of the bottom surface ($p$) as the optimization parameter of position. To prevent potential collisions between the inserted object and existing assets in the original scene, we search for a suitable position around the center of the reconstructed floor. As shown in Fig. 2(b), we first calculate the floor's center $c \leftarrow (c_x, c_y, c_z)$, and set a search square, which uses twice the floor's standard deviation along X axis, $\sigma_x$, and Y axis, $\sigma_y$, as square width and length. The insertion position is sampled from a Uniform distribution inside the search square $p_x \sim \mathcal{U}[c_x - \sigma_x, c_x + \sigma_x]$ and $p_y \sim \mathcal{U}[c_y - \sigma_y, c_y + \sigma_y]$, $p \leftarrow (p_x, p_y, c_z)$. For size ($s$), we use the height of the 3D bounding box of the object as the optimization parameter. For each object category, we first calculate the mean $m_h$ and standard deviation $\sigma_h$ of the height of the object belonging to the same category in the original scene dataset. We then assume the height size follows a Normal distribution and sample a height size from this Normal distribution: $s \in \mathcal{N}(m_h, \sigma_h)$. For the pose ($o$), we only allow the object to rotate along the Z-axis to maintain its stability. The optimization parameter is the rotation angles alone the Z-axis, which follows uniform distribution as $o \sim \mathcal{U}[-\pi, \pi]$.

Algorithm 1 details the Constrained Insertion Parameter Search algorithm. We first set a search budget: $k$ search iterations. For each iteration, we randomly sample each parameter (position, size, and pose) from their corresponding search spaces and calculate the inserted object's bounding box based on the sampled parameters. We then check for collisions with existing objects and quantitatively evaluate the degree of collisions. A direct approach for collision checking is to convert the inserted object into a point cloud and then calculate the overlap with existing objects' point clouds. However, this method is time-consuming due to the large number of points involved. We simplify the problem by converting the original 3D collision into a 2D collision to speed up the collision check. Since the inserted objects are on the floor, if two objects collide, their 3D bounding box projections on the top view would also often collide (but not always, e.g., when an object may be placed under a table; we here ignore these candidate placements). In other words, we disregard the absolute value of the 3D volume and use the 2D collision projection as a relative collision score. Utilizing an efficient collision check allows us to set a relatively large search iteration number, such as $k = 1000$, while still maintaining a limited search time (less than 0.5 seconds). We also consider a resize factor $r$

---

**Algorithm 1:** Constrained Insertion Parameter Search

---

**Input:** An RGBD image of the scene, a reconstructed floor, a 3D object belonging to the class of interest, $j$
**Output:** Position ($\hat{p}$: 3D bounding box bottom center), size ($\hat{s}$: 3D bounding box (bbox) height), and pose
($\hat{o}$: orientation along Z-axis)

---

**1** Compute position search constrains: floor center $c \leftarrow (c_x, c_y, c_z)$, standard deviation $\sigma_x$ and $\sigma_y$

**2** Initialize search parameters: $k \leftarrow 1000$, degree of collision $\hat{l} \leftarrow \inf$

   **for** $i \in \{1, 2, \ldots, k\}$ **do**

**3**      Sample position: $p_x \sim \mathcal{U}[c_x - \sigma_x, c_x + \sigma_x]$ and $p_y \sim \mathcal{U}[c_y - \sigma_y, c_y + \sigma_y]$, $p \leftarrow (p_x, p_y, c_z)$

**4**      Sample size: $s \sim \mathcal{N}(m_h, \sigma_h)$, resize factor $r \sim \mathcal{U}[1, r]$, $s \leftarrow s/r$,
        where $m_h$ and $\sigma_h$ are mean and standard deviation of object height in class $j$ in the raw dataset

**5**      Sample pose: $o \sim \mathcal{U}[-\pi, \pi]$

**6**      Calculate 3D bbox $x_{3D}$, parameter based on the sampled insertion parameter ($p$, $s$ and $o$)

**7**      Project 3D bbox to 2D bbox $x_{2D}$ in top view

**8**      Calculate collision score $l = F(x_{2D})$ with existing objects in the scene

     **if** $l == 0$ **then**
       |  Return $p$, $s$, $o$

     **if** $l < \hat{l}$ **then**
       |  $\hat{p} \leftarrow p$, $\hat{s} \leftarrow s$, $\hat{o} \leftarrow o$
       |  $\hat{l} \leftarrow l$

**9** Return $\hat{p}$, $\hat{s}$, $\hat{o}$

---

to shrink the size of the inserted object to handle inserting a large object in a small empty floor scenario. During the search, we terminate the process if we find an insertion with a collision score of 0; otherwise, we continue to track the best insertion with the lowest collision score and return it after completing $k$ search iterations.

### 3.2 *What* Illumination is on the object

#### 3.2.1 Spatial-varying Illumination Estimation and Retrieval

To answer the question of what kind of illumination should be cast on the object, we first need to estimate the spatially-varying illumination of the scene. This process involves encapsulating intricate global interactions at each spatial location. To achieve this, we utilize the deep inverse rendering framework proposed by Li et al. [2020b]. Initially, we estimate intermediate geometric features such as albedo, normal, depth, and roughness. Subsequently, a LightNet structure, consisting of an encoder-decoder setup, ingests the raw image and the predicted intermediate features. This, in turn, enables the estimation of spatially-varying lighting across the scene.

As depicted in Fig. 2(c), the estimated spatially-varying illumination is represented as environment maps. Specifically, each 4x4 pixel region in the raw image is associated with an environment map, which captures the appearance of the surrounding environment and is used for reflection, refraction, or global illumination. These maps are spherical (equirectangular), representing the environment on a single 2D texture. The X-axis corresponds to longitude, and the Y-axis corresponds to latitude. Each point on the texture corresponds to a specific latitude and longitude on a sphere.

To obtain the environment map associated with the position of the inserted object, we register and retrieve the corresponding environment map based on the estimated position after performing the constrained insertion parameter search.

#### 3.2.2 Environment Map Refinement

**Coordinate transformation.** The environment map, estimated for the inserted object, is based on the local coordinates of the insertion position. In particular, it establishes a coordinate system where the surface normal is designated as the Z-axis. In order to apply this map for relighting the inserted object using a rendering method (such as Blender), it becomes necessary to transform the environment map to align with Blender's coordinate system.

**Latitude completion.** The estimated environment map only contains latitudes in the range $(0, \pi/2)$ because the inverse rendering method cannot estimate the illumination beneath the surface. As shown in Fig. 2(d), we complete the entire environment map by filling in artificial values in the second half.

Table 1: Statistics of external 3D objects from Objaverse [Deitke et al., 2022].

| Category | Bed | Table | Sofa | Chair | Desk | Dresser | Nightstand | Bookshelf | Toilet | Bathtub |
|---|---|---|---|---|---|---|---|---|---|---|
| Number | 190 | 854 | 361 | 934 | 317 | 52 | 13 | 99 | 142 | 24 |

**Intensity refinement.** The estimated environment map is in Low Dynamic Range (LDR) format, lacking High Dynamic Range (HDR) details and high contrast. If we use the predicted value directly, the rendered shadow appears relatively fuzzy. We refine the value by adjusting the scale in log space to estimate the HDR value: $I_{\text{HDR}} = I_{\text{LDR}}^{\gamma}$, where $\gamma$ is a hyperparameter .

Finally, we input the HDR environment map after transformation and refinement, along with the position, size, and pose, into an insertion renderer (e.g., Blender). This allows us to obtain the inserted image with 3D bounding boxes serving as ground truth.

### 3.3 Dataset Augmentation with Insertion and Downstream Model Training

Given an indoor scene dataset and a set of interest classes $\mathcal{C}$ for potential insertion, we can identify external 3D objects set $\mathcal{E}$ that fall within these classes of interest. Before any insertion, we calculate the statistical parameters for each class of interest that we aim to augment. For every class $j \in \mathcal{C}$, we assume the size parameter (for instance, the height) fits a Gaussian distribution. We then calculate the mean and standard deviation of this size parameter to guide the insertion of external objects. Here are the detailed steps for insertion: For each scene within the indoor scene dataset, we randomly select a category $j$ from the class of interest set $\mathcal{C}$. Next, we randomly choose an instance from the external 3D objects set $\mathcal{E}$ that belongs to the selected class $j$. We then utilize our physically plausible insertion method (Algorithm 1) to integrate this external 3D object into the scene. We could train any downstream monocular 3D object detection model with the augmented dataset because we automatically obtain the 3D annotations of the inserted objects.

## 4 Experiments

This section presents experiments to assess the effectiveness of our proposed physically-plausible 3D object insertion method and evaluate how different insertion parameters affect the final performance of monocular 3D object detection.

### 4.1 Dataset and Model Setting

**Indoor scene dataset.** We utilize the SUN RGB-D dataset [Song et al., 2015] as our primary resource for indoor scenes. It is one of the most challenging benchmarks in indoor scene understanding. SUN RGB-D comprises 10,335 RGB-D images captured using four distinct sensors. The dataset is divided into 5,285 training scenes and 5,050 test scenes. Furthermore, it includes 146,617 2D polygons and 58,657 3D bounding boxes, providing a comprehensive dataset for our research.

We also use ScanNet dataset [Dai et al., 2017]. ScanNet v2 is a large-scale RGB-D video dataset, which contains 1,201 videos/scenes in the training set and 312 scenes in the validation set. Adapting it for monocular 3D object detection, we utilized one RGB-D image per video, amounting to 1,201 RGB-D images for training and 312 for validation. We compute the ground truth 3D bounding box label for each of our used views from their provided scene level label, as some objects in the scene may not be visible in our monocular viewpoint.

**External 3D object assets.** The quality of 3D objects is crucial for effective insertion. Hence, we use Objaverse [Deitke et al., 2022], a robust dataset with over 800,000 annotated 3D objects. Using word parsing, we extract objects that align with the classes of interest for monocular 3D object detection within SUN RGB-D. Table 1 shows the selected Objaverse data for each SUN RGB-D class.

**Monocular 3D object detection model.** We focus on the challenging task of monocular 3D object detection that relies solely on a single RGB image as input. We employ ImVoxelNet, which achieves state-of-the-art performance on the raw SUN RGB-D dataset using only a single RGB image as input. Other existing methods either resort to using additional modalities and multiple datasets for extra supervision or exhibit underwhelming performance. For the purpose of monocular 3D object

Table 2: ImVoxelNet 3D monocular object detection performance on the SUN RGB-D dataset with different object insertion methods. When inserting randomly, the accuracy of the downstream object detector drops, i.e., the detector suffers from random insertions (which may have collisions, occlusions, incorrect lighting, etc.). In contrast, by only applying physically plausible position, size, and pose, performance significantly improved (41.80%). Further, when plausible lighting and shadows are added, our 3D copy-paste improves the accuracy of the downstream detector to a new state-of-the-art accuracy (43.79%). We use mAP (%) with 0.25 IOU threshold.

| Setting | Insertion Position, Pose, Size | Insertion Illumination | mAP@0.25 |
|---|---|---|---|
| ImVoxelNet | N/A | N/A | 40.96 |
| ImVoxelNet + random insert | Random | Camera point light | 37.02 |
| ImVoxelNet + 3D Copy-Paste (w/o light) | Plausible position, size, pose | Camera point light | 41.80 |
| ImVoxelNet + 3D Copy-Paste | Plausible position, size, pose | Plausible dynamic light | **43.79** |

Table 3: Per class average precision (AP) of ImVoxelNet 3D monocular object detection performance on SUN RGB-D dataset.

| Setting | mAP@0.25 | bed | chair | sofa | table | bkshf | desk | bathtub | toilet | dresser | nightstand |
|---|---|---|---|---|---|---|---|---|---|---|---|
| ImVoxelNet | 40.96 | 72.0 | 55.6 | 53.0 | 41.1 | **7.6** | 21.5 | 29.6 | 76.7 | 19.0 | 33.4 |
| ImVoxelNet + 3D Copy-Paste | **43.79** | **72.6** | **57.1** | **55.1** | **41.8** | 7.1 | **24.1** | **40.2** | **80.7** | **22.3** | **36.9** |

detection, we train the same ImVoxelNet model on the original SUN RGB-D dataset and its various versions, each augmented via different insertion methods. All mAP results are mAP@0.25.

## 4.2 Physically-plausible position, pose, size, and illumination leads to better monocular detection performance

Our 3D Copy-Paste focuses on solving two challenges: (1) Where and how to put the object: we estimate the object's position, orientation, and size for insertion while ensuring no collisions. (2) What illumination is on the object: we estimate the spatially-varying illumination and apply realistic lighting and shadows to the object rendering. The following experiments evaluate the model performance.

Table 2 presents the results of monocular 3D object detection on the SUN RGB-D dataset, utilizing various object insertion augmentation techniques. The first row is the performance of ImVoxelNet trained on the raw SUN RGB-D dataset without any insertion. The "ImVoxelNet + random insert" row displays results achieved through a naive 3D object insertion without applying physically plausible constraints (random location and Camera point light). This approach led to a drop in accuracy from 40.96% to 37.02%, likely due to the lack of physical plausibility causing severe collisions and occlusions in the final image. The "ImVoxelNet + 3D Copy-Paste (w/o light)" row showcases the performance after implementing our method for only estimating physically plausible insertion position, pose, and size. Despite using a rudimentary camera point light, this approach outperforms "ImVoxelNet" without any insertion, and also outperforms the naive "ImVoxelNet + random insert" (+4.78 % improvement). This result shows that applying plausible geometry is essential for downstream tasks and makes 3D data augmentation useful over a naive, random augmentation. After further applying physically plausible dynamic light, our proposed "ImVoxelNet + 3D Copy-Paste" further improved the performance and achieved new state-of-the-art, surpassing ImVoxelNet without insertion (+2.83 %) on monocular 3D object detection task. This performance improvement suggests that our 3D Copy-Paste insertion can serve as an efficient data augmentation method to positively benefit downstream 3D object detection tasks. Table 3 shows detailed SUN RGB-D monocular 3D object detection results with ImVoxelNet on each individual object category.

Table 4 presents the results of monocular 3D object detection on the ScanNet dataset. We utilized one RGB-D image per video: 1,201 for training and 312 for validation. We compute the ground truth 3D bounding box label for each of our used views from their provided scene-level label. For the baseline, we train an ImVoxelNet monocular 3D object detection model on the training set and test on the validation set. For our method, there are 8 overlapping categories (sofa, bookshelf, chair, table, bed, desk, toilet, bathtub) in the 18 classes of ScanNet with our collected Objaverse data. We use our 3D Copy-Paste to augment the training set and train an ImVoxelNet model. All the training parameters are the same as the training on SUN RGB-D dataset. We show the results on the average accuracy of

Table 4: ImVoxelNet 3D monocular object detection performance on the ScanNet dataset with different object insertion methods.

| Setting | mAP@0.25 | bed | chair | sofa | table | bkshf | desk | bathtub | toilet |
|---|---|---|---|---|---|---|---|---|---|
| ImVoxelNet | 14.1 | 25.7 | 7.9 | **13.2** | 7.8 | 4.2 | 20.5 | 22.1 | **11.5** |
| ImVoxelNet + 3D Copy-Paste | **16.9** | **27.7** | **12.7** | 10.0 | **10.8** | **9.2** | **26.2** | **29.2** | 9.0 |

Table 5: ImVoxelNet 3D monocular object detection performance on SUN RGB-D dataset with different illumination during insertion rendering. All experiments use the same ImVoxelNet model, insertion also uses our proposed physically plausible position, size, and pose.

| Setting | Light source type | Intensity | Direction | With shadow? | mAP@0.25 |
|---|---|---|---|---|---|
| Point Light 1 | Point | 100W | Camera position | Yes | 41.80 |
| Point Light 2 | Point | 100W | Side (left) | Yes | 42.38 |
| Area Light 1 | Area | 100W | Camera position | Yes | 42.67 |
| Area Light 2 | Area | 100W | Side (left) | Yes | 42.02 |
| Spot Light 1 | Spot | 100W | Camera position | Yes | 40.92 |
| Spot Light 2 | Spot | 100W | Side (left) | Yes | 42.10 |
| Sun Light 1 | Sun | 5 | Camera position | Yes | 42.11 |
| Sun Light 2 | Sun | 5 | Side (left) | Yes | 41.21 |
| Ours (Dynamic Light) | Estimated Plausible light | Dynamic | Dynamic | No | 41.83 |
| Ours (Dynamic Light) | Estimated Plausible light | Dynamic | Dynamic | Yes | **43.79** |

the 8 overlapping classes (mAP@0.25) in the Table 4. Our 3D Copy-Paste improves ImVoxelNet by 2.8% mAP.

## 4.3 Ablation study on the influence of insertion illumination and position on monocular 3D object detection

We first explore the influence of illumination of inserted objects on downstream monocular 3D object detection tasks. Table 5 shows the ImVoxelNet performance on SUN RGB-D with different illumination settings during 3D Copy-Paste. To eliminate the influence of other insertion parameters, we fix the estimated position, pose, and size for each scene among all experiments in Table 5.

Fig. 3 provides a visualization of the effects of various light sources and light parameters during the insertion rendering process. The corresponding monocular 3D object detection results are presented in Table 5. These illustrate how lighting not only impacts the visual perception of the inserted object from a human observer's standpoint but also considerably affects the performance of downstream detection tasks. Thus, an accurate and physically plausible lighting estimation is crucial for both understanding the scene and for the practical application of downstream detection tasks.

Table. 2 shows the importance of physical position, pose, and size (local context) on monocular 3D object detection tasks. We also explored the importance of the global context to the detection performance. The global context here means the semantic relationship of the inserted object to the whole scene. For instance, inserting a toilet into a living room may not satisfy the global context. We propose a plausible global context insertion method where the inserted object class considers the global scene information. Also, we could select an inserted class based on the floor size: insert larger size objects (e.g., bed, bookshelf) on only a large size floor. Table. 6 shows results on different settings. We find considering the global context during the insertion is on par with the random category selecting setting, and the following downstream detection model may not be sensitive to that.

## 4.4 Qualitative Analysis

Fig. 4 shows the qualitative results of monocular 3D object detection on SUN RGB-D dataset. Our method demonstrates enhanced capabilities in detecting objects with significant occlusion, provides improved pose estimation, and effectively suppresses false positives.

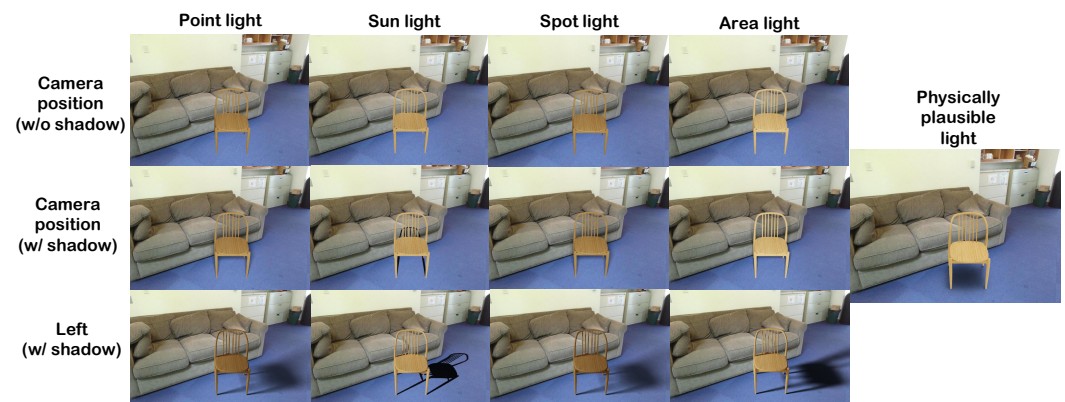

Figure 3: Visualization of different illumination on inserted objects.

Table 6: Ablation study of global context influence on ImVoxelNet monocular 3D object detection performance on SUN RGB-D.

| Method | Follow global context? | Select class based on empty size? | mAP@0.25 |
|---|---|---|---|
| ImVoxelNet + 3D Copy-Paste | Yes | No | 43.75 |
| ImVoxelNet + 3D Copy-Paste | Yes | Yes | 43.74 |
| ImVoxelNet + 3D Copy-Paste | No | Yes | 42.50 |
| ImVoxelNet + 3D Copy-Paste | No | No | **43.79** |

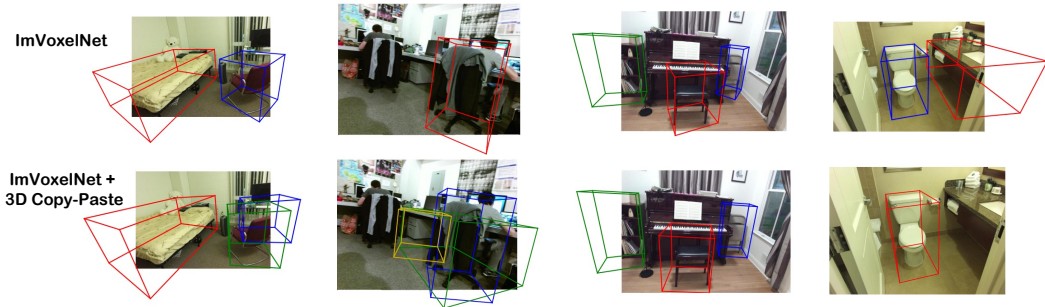

Figure 4: Qualitative results on the SUN RGB-D dataset.

## 5   Conclusion and Discussion

Our work addresses the challenge of scarce large-scale annotated datasets for monocular 3D object detection by proposing a physically plausible indoor 3D object insertion approach. This technique allows us to effectively augment existing indoor scene datasets, such as SUN RGB-D, with large-scale annotated 3D objects that have both plausible physical location and illumination. The resulting augmented dataset enables training a monocular 3D object model that achieves new state-of-the-art performance. Our approach carefully considers physically feasible locations, sizes, and poses for inserted objects, avoiding collisions with the existing room layout, and estimates spatially-varying illumination to seamlessly integrate the objects into the original scene. We also systematically evaluate the impact of the physical position and illumination of the inserted objects on the performance of the final monocular 3D object detection model. This paper is the first to demonstrate that physically plausible 3D object insertion can serve as an effective generative data augmentation technique, leading to state-of-the-art performance in discriminative downstream tasks like monocular 3D object detection. Our findings highlight the potential of 3D data augmentation in improving the performance of 3D perception tasks, opening up new avenues for research and practical applications.

**Acknowledgments.** This work is in part supported by Bosch, Ford, ONR MURI N00014-22-1-2740, NSF CCRI #2120095, Amazon ML Ph.D. Fellowship, National Science Foundation (award 2318101), C-BRIC (one of six centers in JUMP, a Semiconductor Research Corporation (SRC) program sponsored by DARPA) and the Army Research Office (W911NF2020053). The authors affirm that the views expressed herein are solely their own, and do not represent the views of the United States government or any agency thereof.

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

# A   Experiments on more Monocular 3D Object Detection methods

In our main paper, we utilize ImVoxelNet [Rukhovich et al., 2022] for monocular 3D object detection. To show the robustness of our 3D Copy-Paste across different downstream detection methods. We conducted additional experiments with another monocular 3D object detection model: Implicit3DUnderstanding (Im3D [Zhang et al., 2021]). The Im3D model predicts object 3D shapes, bounding boxes, and scene layout within a unified pipeline. Training this model necessitates not only the SUN RGB-D dataset but also the Pix3D dataset [Sun et al., 2018], which supplies 3D mesh supervision. The Im3D training process consists of two stages. In stage one, individual modules - the Layout Estimation Network, Object Detection Network, Local Implicit Embedding Network, and Scene Graph Convolutional Network - are pretrained separately. In stage two, all these modules undergo joint training. We incorporate our 3D Copy-Paste method only during this second stage of joint training, and it's exclusively applied to the 10 SUN RGB-D categories we used in the main paper. We implemented our experiment following the official Im3D guidelines[1].

Table 7 displays the Im3D results for monocular 3D object detection on the SUN RGB-D dataset, adhering to the same ten categories outlined in main paper. Im3D without insertion, attained a mean average precision (mAP) detection performance of 42.13%. After applying our 3D Copy-Paste method—which encompasses physically plausible insertion of position, pose, size, and light—the monocular 3D object detection mAP performance increased to 43.34. These results further substantiate the robustness and effectiveness of our proposed method.

Table 7: Im3D [Zhang et al., 2021] 3D monocular object detection performance on the SUN RGB-D dataset (same 10 categories as the main paper).

| Setting | Insertion Position, Pose, Size | Insertion Illumination | mAP |
|---|---|---|---|
| Im3D | N/A | N/A | 42.13 |
| Im3D + 3D Copy-Paste | Plausible position, size, pose | Plausible dynamic light | **43.34** |

# B   More experiment details

We run the same experiments multiple times with different random seeds. Table 8 shows the main paper Table 2 results with error range.

Table 8: ImVoxelNet 3D monocular object detection performance on the SUN RGB-D dataset with different object insertion methods (with error range).

| Setting | Insertion Position, Pose, Size | Insertion Illumination | mAP@0.25 |
|---|---|---|---|
| ImVoxelNet | N/A | N/A | $40.96 \pm 0.4$ |
| ImVoxelNet + random insert | Random | Camera point light | $37.02 \pm 0.4$ |
| ImVoxelNet + 3D Copy-Paste (w/o light) | Plausible position, size, pose | Camera point light | $41.80 \pm 0.3$ |
| ImVoxelNet + 3D Copy-Paste | Plausible position, size, pose | Plausible dynamic light | $\mathbf{43.79} \pm 0.4$ |

We also show our results with mAP@0.15 on SUN RGB-D dataset (Table 9), our method shows consistent improvements.

Table 9: ImVoxelNet 3D monocular object detection performance on the SUN RGB-D dataset with mAP@0.15.

| Setting | Insertion Position, Pose, Size | Insertion Illumination | mAP@0.15 |
|---|---|---|---|
| ImVoxelNet | N/A | N/A | 48.45 |
| ImVoxelNet + 3D Copy-Paste | Plausible position, size, pose | Plausible dynamic light | **51.16** |

---

[1]https://github.com/chengzhag/Implicit3DUnderstanding

# C Discussion on Limitations and Broader Impact

**Limitations.** Our method, while effective, does have certain limitations. A key constraint is its reliance on the availability of external 3D objects, particularly for uncommon categories where sufficient 3D assets may not be readily available. This limitation could potentially impact the performance of downstream tasks. Moreover, the quality of inserted objects can also affect the results. Possible strategies to address this limitation could include leveraging techniques like Neural Radiance Fields (NeRF) to construct higher-quality 3D assets for different categories.

**Broader Impact.** Our proposed 3D Copy-Paste method demonstrate that physically plausible 3D object insertion can serve as an effective generative data augmentation technique, leading to state-of-the-art performance in discriminative downstream tasks like monocular 3D object detection. The implications of this work are profound for both the computer graphics and computer vision communities. From a graphics perspective, our method demonstrates that more accurate 3D property estimation, reconstruction, and inverse rendering techniques can generate more plausible 3D assets and better scene understanding. These assets not only look visually compelling but can also effectively contribute to downstream computer vision tasks. From a computer vision perspective, it encourages us to utilize synthetic data more effectively to tackle challenges in downstream fields, including computer vision and robotics.

