# OpenReview forum: "3D Copy-Paste: Physically Plausible Object Insertion for Monocular 3D Detection"
_NeurIPS.cc/2023/Conference — NeurIPS 2023 poster_

### Official Review · Reviewer_mHBD · 2023-07-04

**Soundness:** 3 good
**Presentation:** 2 fair
**Contribution:** 2 fair
**Rating:** 5
**Confidence:** 5

**Summary:**

This paper proposes a data augmentation approach to assisting the training of monocular 3D detectors by inserting 3D objects into indoor scenes in a physically plausible manner. Specifically, it addresses two main challenges in the entire pipeline: 1) where and how to insert those objects; 2) which illumination on the object makes the rendering photorealistic. Experiments validate that the data generated by the overall pipeline can enhance the state-of-the-art monocular 3D detectors. Detailed ablation studies further provide some insights regarding which aspects are important in the proposed method.

**Strengths:**

- The basic idea is easy to follow and the illustration figures are clear.
- The overall pipeline is proven effective for enhancing downstream 3D perception systems.
- This pipeline is systematic and comprehensive, taking almost most of the aspects when inserting 3D objects into scenes and image data generation for training monocular detectors.
- The key insights regarding three challenges in the introduction and two critical considerations (where and how & illumination) in the methodology are accurate.
- The proposed method achieves new state-of-the-art on the SUN RGB-D benchmark and has detailed ablation studies to reveal which aspects are most essential in the overall pipeline. (For example, the analysis in Table 4 is interesting.)

**Weaknesses:**

- (Related work) The related work section has many inaccurate statements, such as MV3D is a multi-view method incorporating both LiDAR-based point clouds and images and VoxelNet is a LiDAR-only method, which should not be discussed in the monocular 3D detection section. In contrast, there are many missing works of monocular 3D detection in driving scenarios, such as 3DOP[1], MLFusion[2], M3D-RPN[3], MonoDIS[4], Pseudo-LiDAR[5], FCOS3D[6], SMOKE[7], RTM3D[8], PGD[9], CaDDN[10], etc. There is also some missing literature for 3D Data Augmentation, such as MoCa[11], GeoAug[12], etc.

- (Methodology) The overall pipeline is the main contribution of this paper. From another perspective, the most important part is to combine different "existing" techniques together and make it finally work to produce photorealistic images after inserting 3D objects. The key flaw here is that most of the techniques used in each stage are not newly proposed in this work, making this work more like an engineering one. (although I admit that the overall pipeline is still a good contribution to the community)

- (Experiments) The proposed method is not limited to any method and any dataset, but only tested on ImVoxelNet and SUN RGB-D. Although it can demonstrate the basic effectiveness, it would be much more convincing if the author can provide more ablation results on other baselines and datasets.

[1] 3D Object Proposals for Accurate Object Class Detection

[2] Multi-Level Fusion Based 3D Object Detection from Monocular Images

[3] M3D-RPN: Monocular 3D Region Proposal Network for Object Detection

[4] Disentangling Monocular 3D Object Detection

[5] Pseudo-LiDAR from Visual Depth Estimation: Bridging the Gap in 3D Object Detection for Autonomous Driving

[6] FCOS3D: Fully Convolutional One-Stage Monocular 3D Object Detection

[7] SMOKE: Single-Stage Monocular 3D Object Detection via Keypoint Estimation

[8] RTM3D: Real-time Monocular 3D Detection from Object Keypoints for Autonomous Driving

[9] Probabilistic and Geometric Depth: Detecting Objects in Perspective

[10] Categorical Depth Distribution Network for Monocular 3D Object Detection

[11] Exploring Data Augmentation for Multi-Modality 3D Object Detection

[12] Exploring Geometric Consistency for Monocular 3D Object Detection

**Questions:**

None.

**Limitations:**

The author discusses the limitations and social impacts in the supplementary material.

---

> ### Author Rebuttal · Authors · 2023-08-09
>
> Thank you for your time and comments! Please see our response below.
>
> - **Related work**:
> Thanks for your suggestion and comments! We will modify the statement and add all the provided related papers.
>
>
> - **Methodology**:
> To conduct 3D object insertion for data augmentation, traditional methods [1] require manually locating support planes, designing poses, estimating lighting, etc, which are hard to scale up. Out method is the first automated 3D insertion pipeline on complex indoor scenes, enabling large-scale 3D insertion for data augmentation.
>
>   To allow a fully automated pipeline, we make the following technical contributions: (1) To avoid collision, after plane detection, we propose a constrained insertion parameter search method (algorithm 1) to guarantee a physically-plausible inserted position, pose, and size. We also include an efficient collision check method to solve the time-consuming challenge. (2) We conduct environment map transformation and refinement to add more accurate illumination on inserted objects.
>
>   To our best knowledge, our work is the first to showcase that physically plausible 3D object insertion can serve as an effective generative data augmentation technique for indoor scenes, leading to state-of-the-art performance in discriminative downstream tasks such as monocular 3D object detection, opening up new avenues for research and practical applications.
>
> - **More experiments on other models and datasets**:
>
>   **For more monocular 3D object detection methods**, we conducted experiments on Implicit3DUnderstanding (Im3D[2]) in supplementary materials, and the results are in supplementary Table S1. Using our method (Im3D + 3D Copy-Paste) improve the mAP_0.25 from 42.13 (Im3D) to 43.34.
>
>   **For more datasets**, we extend our method to ScanNet [3] dataset. Here are the detailed setting and results on ScanNet:
>
>   [*Adapt to monocular setting*] ScanNet contains 1,201 videos(scene) in the training set and 312 videos(scene) in the validation set. For monocular 3D object detection, we use one RGB-D image per video, so 1,201 RGB-D images for training and 312 for validation (test). We calculate the ground truth 3D bounding box label for each of our used views from their provided video(scene) level label (because some objects in the scene may not be visible in our monocular view).
>
>   [*Training and test*] For the baseline, we train an ImVoxelNet monocular 3D object detection model on the training set and test on the validation set. For our method, there are 8 overlap categories (sofa, bookshelf, chair, table, bed, desk, toilet, bathtub) in the 18 classes of ScanNet with our collected Objaverse data (main paper Table1). We use our 3D copy-past to augment the training set and train an ImVoxelNet. All the training parameters are the same as the training on SUN RGB-D dataset. We will release all the code.
>
>     We show the results on the average accuracy of the 8 overlap classes (AP_0.25) in the Table below. Our 3D Copy-Paste improves ImVoxelNet by 2.8% mAP.
>
> |ScanNet AP_0.25                   | Average (mAP)|bed |chair|sofa|table|bookshelf|desk|bathtub|toilet|
> |-----------------|------|------|----|----|----|----|----|----|----|
> | ImVoxelNet              |14.1 |25.7|7.9  |**13.2**|7.8  |4.2      |20.5|22.1   |**11.5**  |
> | ImVoxelNet+3D Copy-Paste|**16.9** |**27.7**|**12.7** |10.0|**10.8** |**9.2**      |**26.2**|**29.2**   |9.0   |
>
>
>
>
> **References**
>
> [1] Debevec, P., 2008. Rendering synthetic objects into real scenes: Bridging traditional and image-based graphics with global illumination and high dynamic range photography. In Acm siggraph 2008 classes (pp. 1-10).
>
> [2] Zhang, C., Cui, Z., Zhang, Y., Zeng, B., Pollefeys, M. and Liu, S., 2021. Holistic 3d scene understanding from a single image with implicit representation. In Proceedings of the IEEE/CVF Conference on Computer Vision and Pattern Recognition (pp. 8833-8842).
>
> [3] Dai, A., Chang, A.X., Savva, M., Halber, M., Funkhouser, T. and Nießner, M., 2017. Scannet: Richly-annotated 3d reconstructions of indoor scenes. In Proceedings of the IEEE conference on computer vision and pattern recognition (pp. 5828-5839).

---

> > ### Comment · Reviewer_mHBD · 2023-08-15
> > **Response to Rebuttal**
> >
> > I acknowledge that I have read the authors' rebuttal and the other reviews.
> >
> > Thanks for your rebuttal and I feel most of my concerns are addressed. Given my impression that this paper has done solid work but may have relatively small technical contributions, I will keep my score in the final decision but support its acceptance considering its value to this community.

---

> > > ### Author Response · Authors · 2023-08-21
> > > **Thank you for updating your review!**
> > >
> > > Thank you for your feedback! We will incorporate the suggested changes in the revised paper.

---

### Official Review · Reviewer_B4nT · 2023-07-06

**Soundness:** 3 good
**Presentation:** 3 good
**Contribution:** 3 good
**Rating:** 6
**Confidence:** 3

**Summary:**

This work introduces 3D Copy-Paste, a  physically plausible indoor object insertion technique for automatically generating large-scale annotated 3D objects. This approach ensures the plausibility of the objects’ physical location, size, pose, and illumination within the scene
Using this 3D copy-paste as augmentation, a better monocular 3D detector can be trained.
Experiments are demonstrate on the SUN RGB-D dataset, demonstrating the effectiveness of the proposed object insertion method in improving the 3D detection.


**Strengths:**

1. The method is clearly explained
2. The proposed methods for object insertion can help create annotated data for free.
3. The inserted object is physically plausible in location, size and illumination. Moreover, it's fully automated.
4. Solid experiments are conducted in SUN RGB-D based on a SOTA detector (ImVoxelNet), and the detection AP is improved using the proposed method as augmentation. (40.96-->43.79)

**Weaknesses:**

This work may somewhat lack novelty. I'm not familiar with the indoor scene object insertion task, but for the outdoor scene, there are quite a lot similar works[1-4].  It would be better to include these works in the literature review. These works share similar ideas with the authors, where they insert object in physically plausible location and illumination. And some works also test with downstream tasks and demonstrate effectiveness of using actor insertion as data augmentation.    Besides,  I'm not fully convinced by the claim that indoor scenes are "more challenging" in L51. The layout, illumination, complexity of outdoor scenes are apparently more complex.

[1] Augmented Reality Meets Computer Vision : Efficient Data Generation for Urban Driving Scenes
[2] GeoSim: Realistic Video Simulation via Geometry-Aware Composition for Self-Driving
[3] Neural Light Field Estimation for Street Scenes with Differentiable Virtual Object Insertion
[4] 3D Data Augmentation for Driving Scenes on Camera


**Questions:**

I'm not familiar with the ImVoxelNet,I just quickly went through it to review this paper.  ImVoxelNet reported AP@0.15. Why AP@0.25 is reported in this paper?

I will raise my rating if the authors can apply this approach to outdoor scenes  like kitti and nuscenes  and improve ImVoxelNet results.(ImVoxelNet is already tested on this dataset)


**Limitations:**

Apparently, the authors did not go over https://neurips.cc/public/guides/PaperChecklist.
Not many experiments and implemented details are provided, Licenses and Assets  are not clearly described either.
But I don't penalize it in the rating though.

---

> ### Author Rebuttal · Authors · 2023-08-09
>
> Thank you for your time and comments! Please see our response below.
>
> - **Indoor/outdoor difference and novelty**:
> Different from outdoor scenes, the challenges in indoor scenes include (1) complex spatial layouts (in particular, cluttered background and limited object-placeable space) that necessitate a carefully-designed method to allow automated object placement (physically-plausible position, size, pose), and (2) complex lighting effects such as soft shadow, inter-reflection and long-range light source dependency that demand dealing with lighting for harmonious object insertion.
>
>   To deal with the above challenges, we make the following key technical contributions: (1) To avoid collision, after plane detection, we propose constrained insertion parameter search (algorithm 1) to guarantee a physically-plausible inserted position, pose and size. We also include an efficient collision check method to solve the time-consuming challenge. (2) We conduct environment map transformation and refinement to add more accurate illumination on inserted objects.
>
>   To our best knowledge, our work is the first automated 3D insertion pipeline on complex indoor scenes, enabling large-scale 3D insertion for data augmentation. It also showcases that physically plausible 3D object insertion can serve as an effective generative data augmentation technique for indoor scenes, leading to state-of-the-art performance in discriminative downstream tasks such as indoor monocular 3D object detection, opening up new avenues for research and practical applications.
>
>   Thank you for your suggestion, we will modify the manuscript as follows:
>   - we already cite your ref [3] Neural Light Field Estimation but will add the other outdoor citations.
>   - we will tone down the claim that indoor scene is more challenging since it may indeed be hard to defend.
>   - we will extend our discussion of how our work compares to previous art with outdoor scenes.
>
>
> - **AP_0.25 and AP_0.15**:
> This has to do with the fact that official results are only available for a few combinations of dataset setting, object classes, and IOU threshold: The official ImVoxelNet GitHub code, when using SUN RBG-D on “10 classes from VoteNet” setting (same as ours), also uses 0.25 IOU threshold with performance 40.7 (even though they use 0.15 with other datasets/classes).
> The authors provided an implementation in MMdetection3D [1] Github, which also uses mAP 0.25 for the 10 classes from VoteNet setting (performance 40.96). We used the official code of MMdetection3D, so we used the 0.25 IOU threshold. Per your suggestion, we also show our results with mAP 0.15 on SUN RGB-D dataset (10 classes from VoteNet) below, our method shows consistent improvements.
>
> |SUN RGB-D        | mAP_0.15|
> |-----------------|------|
> | ImVoxelNet              |48.45 |
> | ImVoxelNet + 3D Copy-Paste|**51.16**|
>
>
> - **More experiments on outdoor dataset**:
> Thanks! In this paper, we focus on indoor scene insertion, we will treat extension to the outdoor scene as a future work to explore. In the meantime, we note that our title and overall positioning of the paper make it very clear that it is currently focused on indoor scenes (i.e., we are not overselling the work).
>
>   However, we added experiments on the new dataset ScanNet[2] and new models (Implicit3DUnderstanding[3] in supplementary) as a way to show the generalization of our method. Here are the detailed setting and results on ScanNet:
>
>   [*Adapt to monocular setting*] ScanNet contains 1,201 videos(scene) in the training set and 312 videos(scene) in the validation set. For monocular 3D object detection, we use one RGB-D image per video, so 1,201 RGB-D images for training and 312 for validation (test). We calculate the ground truth 3D bounding box label for each of our used views from their provided video(scene) level label (because some objects in the scene may not be visible in our monocular view).
>
>   [*Training and test*] For the baseline, we train an ImVoxelNet monocular 3D object detection model on the training set and test on the validation set. For our method, there are 8 overlap categories (sofa, bookshelf, chair, table, bed, desk, toilet, bathtub) in the 18 classes of ScanNet with our collected Objaverse data (main paper Table1). We use our 3D copy-past to augment the training set and train an ImVoxelNet. All the training parameters are the same as the training on SUN RGB-D dataset. We will release all the code and training data.
>
>   We show the results on the average accuracy of the 8 overlap classes (AP_0.25) in the Table below. Our 3D Copy-Paste improves ImVoxelNet by 2.8% mAP.
>
> |ScanNet AP_0.25  | Average (mAP)|bed |chair|sofa|table|bookshelf|desk|bathtub|toilet|
> |---|--|--|--|----|----|----|----|----|----|
> | ImVoxelNet              |14.1 |25.7|7.9  |**13.2**|7.8  |4.2      |20.5|22.1   |**11.5**  |
> | ImVoxelNet+3D Copy-Paste|**16.9** |**27.7**|**12.7** |10.0|**10.8** |**9.2**      |**26.2**|**29.2**   |9.0   |
>
>
> - **Paper checklist**:
> Thanks for your reminder! The limitations and broader impact discussion were in the supplementary, we also added more experiment details in the supplementary. We will add more experiment and implementation details and corresponding Licenses and Assets, and double-check the PaperChecklist.
>
> **References**
>
> [1] Contributors, M. (2020). MMDetection3D: OpenMMLab next-generation platform for general 3D object detection.
>
> [2] Dai, A., Chang, A.X., Savva, M., Halber, M., Funkhouser, T. and Nießner, M., 2017. Scannet: Richly-annotated 3d reconstructions of indoor scenes. In Proceedings of the IEEE conference on computer vision and pattern recognition (pp. 5828-5839).
>
> [3] Zhang, C., Cui, Z., Zhang, Y., Zeng, B., Pollefeys, M. and Liu, S., 2021. Holistic 3d scene understanding from a single image with implicit representation. In Proceedings of the IEEE/CVF Conference on Computer Vision and Pattern Recognition (pp. 8833-8842).

---

> > ### Comment · Reviewer_B4nT · 2023-08-19
> > **Thanks for the response**
> >
> > Thank you for the response, especially on the new metric setting and results on ScanNet.
> >
> > After reading others reviews and response from author, I plan to maintain weak accept decision. Overall this work is decent without significant flaws.
> > The reasons preventing me from raising rating are:
> > 1. No experiments on outdoor scenes, which is the biggest consideration for me to raising score before rebuttal.
> > 2. I hold similar ideas to 7hAU on that it may not be necessary to build such a complicated system, just to improve the mAP a bit, though I understand that it's hard to to  achieve such improvements.  One suggestion for the work is that to also take realism as consideration, and may add some non-paired metrics like FID and human evaluation.
> >
> > Best

---

> > > ### Author Response · Authors · 2023-08-21
> > > **Thank you for your feedback!**
> > >
> > > Thank you for your feedback!
> > >
> > > We agree with the importance of experimenting with outdoor scenes and will prioritize it in our future work.
> > >
> > > We agree with the study on realism. In fact, our main paper's Table 3 presents preliminary results on this matter. As lighting and shadow play pivotal roles in photorealism, we investigated the effects of various lighting scenarios and the presence or absence of shadows on monocular 3D object detection outcomes. Our findings indicate that a more photorealistic insertion, characterized by the use of physically plausible lighting and the inclusion of shadows, tends to enhance downstream detection performance.

---

### Official Review · Reviewer_DWmW · 2023-07-06

**Soundness:** 4 excellent
**Presentation:** 4 excellent
**Contribution:** 3 good
**Rating:** 5
**Confidence:** 4

**Summary:**

This paper presents a novel 3D augmentation method to augment the variety of 3D scenes with the corresponding 2D images. This augmentation method focuses on addressing the data-hungry problem when doing monocular 3D detection. The strategy here is intuitive: to insert 3D synthetic object models into 3D real scenes to augment the 3D scene data, and make sure consistent illumination, shading, and layout reasonability without any object collision issues. Therefore, the pipeline in this paper consists of three parts to answer three questions: 1. where and how to place the object in a 3D scene; 2. what is the illumination and how to add it onto object; 3. use the augmented dataset for network training.

In my view, the major contribution of this paper is the pipeline or concept: to leverage inverse rendering and re-rendering to augment 3D data and its corresponding 2D image for monocular 3D detection. Each module used in the pipeline already exists.

The experiments in this paper are pretty extensive, and this paper is well written.



**Strengths:**

As discussed above, the major strength of this paper is its concept and the tailored pipeline.

1. This paper proposed a new strategy to do 2D-3D data augmentation using neural inverse rendering and re-rendering.

2. The tailored pipeline successfully verified the idea, that such a data augmentation strategy can improve monocular 3D detection by a large margin.


**Weaknesses:**

In my view, the weakness is from the method contribution side:

1. Neural inverse rendering to support object editing and augmenting is not novel. There are many works in image-based rendering and decomposition that can support inserting new objects into a 3D scene. I understand that the contribution here is to use it for data augmentation to support monocular 3D detection. But the methodology here is not novel.

2. Many modules are from existing works (e.g., the inverse rendering framework, plane extraction) which makes this paper more like a novel combination to improve an existing task. I like the insight here by leveraging inverse rendering to do data augmentation for monocular 3D detection, and the performance is quite good. It would be more convincing if it also works for other datasets, e.g., ARKitScenes. Because in my view, the experiment performance is the other major contribution.

**Questions:**

1. I wonder how to choose the object class category for insertion. Is it randomly sampled from an object class set? or manually? Because for indoor scenes, there is a strong scene context, e.g., it is not much likely to insert a "bed" in a "toilet". How do you consider such consistency?

**Limitations:**

The authors discussed the limitation and societal impact in the supplemental.

---

> ### Author Rebuttal · Authors · 2023-08-09
>
> Thank you for your time and comments! Please see our response below.
>
> - **Methodology contribution**:
> To conduct 3D object insertion for data augmentation, traditional methods[1] require manually locating support planes, designing poses, estimating lighting, etc, which are hard to scale up. Out method is the first automated 3D insertion pipeline on complex indoor scenes, enabling large-scale 3D insertion for data augmentation.
>
>   To allow a fully automated pipeline, we make the following technical contributions: (1) To avoid collision, after plane detection, we propose a constrained insertion parameter search method (algorithm 1) to guarantee a physically-plausible inserted position, pose, and size. We also include an efficient collision check method to solve the time-consuming challenge.  (2) We conduct environment map transformation and refinement to add more accurate illumination on inserted objects.
>
>   To our best knowledge, our work is the first to showcase that physically plausible 3D object insertion can serve as an effective generative data augmentation technique for indoor scenes, leading to state-of-the-art performance in discriminative downstream tasks such as monocular 3D object detection, opening up new avenues for research and practical applications.
>
>
>
> - **Experiments on other datasets**:
> Thank you for your suggestion. We conduct new experiments on the ScanNet [2] dataset. Here are the detailed setting and results:
>
>   [*Adapt to monocular setting*] ScanNet contains 1,201 videos(scene) in the training set and 312 videos(scene) in the validation set. For monocular 3D object detection, we use one RGB-D image per video, so 1,201 RGB-D images for training and 312 for validation (test). We calculate the ground truth 3D bounding box label for each of our used views from their provided video(scene) level label (because some objects in the scene may not be visible in our monocular view).
>
>   [*Training and test*] For the baseline, we train an ImVoxelNet monocular 3D object detection model on the training set and test on the validation set. For our method, there are 8 overlap categories (sofa, bookshelf, chair, table, bed, desk, toilet, bathtub) in the 18 classes of ScanNet with our collected Objaverse data (main paper Table1). We use our 3D copy-past to augment the training set and train an ImVoxelNet. All the training parameters are the same as the training on SUN RGB-D dataset. We will release all the code.
>
>   We show the results on the average accuracy of the 8 overlap classes (AP_0.25) in the Table below. Our 3D Copy-Paste improves ImVoxelNet by 2.8% mAP.
>
> |ScanNet AP_0.25                   | Average (mAP)|bed |chair|sofa|table|bookshelf|desk|bathtub|toilet|
> |-----------------|------|------|----|----|----|----|----|----|----|
> | ImVoxelNet              |14.1 |25.7|7.9  |**13.2**|7.8  |4.2      |20.5|22.1   |**11.5**  |
> | ImVoxelNet+3D Copy-Paste|**16.9** |**27.7**|**12.7** |10.0|**10.8** |**9.2**      |**26.2**|**29.2**   |9.0   |
>
>
>
> - **Inserted object class selection**:
> Good point! The main results in Tables 2 and 3 use a random sample from the object class set, taken uniformly and without consideration of context. However, we also explore the influence of global context on detection performance in the main paper, Table 4. For this experiment, we only insert the object categories already existing in the current room to make the insertion more globally semantically plausible (e.g., avoid inserting toilets into other rooms except for the bathroom). For instance, if the room contains a table, chair, and sofa, we only consider inserting new objects that belong to these 3 categories.
>
>   The results (Table 4) show that considering the global semantic meaning (43.75) is on par with the random category selecting setting (43.79). One potential reason is that the downstream detection (CNN-based models) may rely more on local information to conduct detection, so they are not sensitive to the global semantics. Different from point cloud-based 3D detection, where context information is important as RGB information is often discarded, in monocular 3D object detection where the input is an RGB image, appearance per se may be the most important clue.
>
>
> **References**
>
> [1] Debevec, P., 2008. Rendering synthetic objects into real scenes: Bridging traditional and image-based graphics with global illumination and high dynamic range photography. In Acm siggraph 2008 classes (pp. 1-10).
>
> [2] Dai, A., Chang, A.X., Savva, M., Halber, M., Funkhouser, T. and Nießner, M., 2017. Scannet: Richly-annotated 3d reconstructions of indoor scenes. In Proceedings of the IEEE conference on computer vision and pattern recognition (pp. 5828-5839).

---

> > ### Comment · Reviewer_DWmW · 2023-08-17
> >
> > Thanks to the authors for their detailed response. After thoroughly reading, I would like to raise my score to "weak accept", and I would strongly suggest the authors include the new experiments in the main paper.
> >
> > For your second response on "Inserted object class selection", I agree that the 2D detection method relies on CNNs that prioritize local information and it is view-dependent, but it is hard to judge if the point-based method relies on global clues more than the local ones or not. It depends on what backbone you use. I highlight the class selection here since I would like to know if the augmented dataset shares the same class distribution with the original distribution, or if it improves/balances the original class distribution to make your method work for long-tail classes.

---

> > > ### Author Response · Authors · 2023-08-18
> > > **Thank you for your response!**
> > >
> > > Thank you for your response! Yes, we will include the new experiments in the revised paper.
> > >
> > > - **Backbone influence**
> > > Yes, we agree that the backbone is important for the point-based method.
> > >
> > > - **Class selection**
> > > For the insertion, yes, and we tried both in main paper table 4, where “Follow global context‘’ represent relatively keeping the original distribution and “Not Follow global context” may balance the original class distribution. We did not observe significant differences.

---

### Official Review · Reviewer_sM4b · 2023-07-06

**Soundness:** 3 good
**Presentation:** 3 good
**Contribution:** 3 good
**Rating:** 6
**Confidence:** 4

**Summary:**

This paper addresses the scarcity of large-scale annotated datasets which is challenging for rapid deployment of monocular 3D object detection models. A physically plausible indoor 3D object insertion approach is proposed to automatically copy and paste virtual objects into real scenes. The resulting objects have 3D bounding boxes with plausible physical locations and illumination, augmenting existing indoor scene datasets. The proposed data augmentation method achieves state-of-the-art performance in monocular 3D object detection. Most importantly, in this approach, a candidate selection process is applied along with a spatially varying illumination procedure from an existing method to ensure the plausibility of the objects’ physical location, size, pose, and illumination within the scene.  From the results, the location and illumination of the inserted objects appear to have a significant impact on the performance of the downstream model.

**Strengths:**

This paper's proposed approach is impressive. It automatically inserts virtual objects from ObjaVerse into real scenes, addressing the issue of limited annotated datasets in computer vision. The method ensures the objects' physical location, size, pose, and illumination, resulting in augmented indoor scene datasets. The use of plane selection methods, discarding objects with collisions using a simplified assumption, and constrained parameter search for insertion, along with the use of spatially-varying lighting estimation is well-thought-of and designed process. Its improved performance from resulting augmentations for monocular 3D object detection demonstrates the effectiveness of the method. Furthermore, the paper is very well-written and easy to follow.

**Weaknesses:**

The paper lacks a comparison or discussion in relation to Common 3D Corruption (CVPR 2022; not cited). Even though common 3d corruption only evaluates 2D downstream tasks, it could still be utilized and compared for 3D object detection and demonstrate how important is 3D and illumination-aware physically grounded insertions. Another weakness is that the paper only evaluates one task. It would be beneficial to assess the impact of 3D grounded and illumination-aware object insertion on other 3D or 2D tasks. Could it also enhance 2D recognition tasks such as segmentation or detection? Including these aspects in the paper would reinforce its findings. It appears that the paper is lacking a basic 2D copy-paste baseline as well. It would be interesting to see how much this technique could potentially improve 3D tasks (Q: Is random insert in comparisons a 2D insertion or 3D insertion?). Additionally, the paper is missing ablation results for changes in object detection performance for each individual object.

**Questions:**

I have a few questions that I hope the authors can help with.

- First, I'm curious about what happens to out-of-distribution insertions such as the NYU dataset. Would it be helpful to include more datasets?
- Second, I'm wondering how long it takes to render one object, taking into account the time it takes to search for the insertion point, correct the appearance, and other factors.
- Lastly, I've noticed that most of the inserted objects are generally diffuse. I'm curious about how the object insertion process is affected when inserting shiny or specular objects.


**Limitations:**

Missing comparison with common 3D corruptions as data augmentation and comparing 2D recognition tasks + other 3D downstream tasks.

---

> ### Author Rebuttal · Authors · 2023-08-09
>
> Thank you for your time and comments! Please see our response below.
>
> - **Comparison to Common 3D Corruption**:
> 3D Common corruptions (3DCC) use 3D information to generate real-world corruptions, which can evaluate the model robustness and be used as a data augmentation for model training. Our method's contribution may be orthogonal to 3DCC. 3DCC conducted scene-level global augmentation and did not introduce new object content. Combining our method with 3DCC may achieve better performance. We will cite this paper and add comparisons in related works.
>
>
> - **Evaluation on other tasks**:
> That is a very good point, we focus on monocular 3D object detection because it is a challenging, fundamentally representative, and data-intensive task, which involves both 3D geometry and semantic understanding, and our method could help. We also conducted experiments to show that the inserted position, size, pose, and lighting do influence the downstream model performance. We extend our method in other 3D detection datasets (ScanNet [1] below) and other models (Implicit3DUnderstanding [2] in supplementary), and we will treat extending to other tasks as future work.
>
>   While we agree with the reviewer that extending to other downstream tasks is desirable, this will take more time than available during the rebuttal period; in the meantime, we note that our title and overall positioning of the paper are not overselling our results, i.e., it is clearly stated that our method is useful for monocular 3D object detection. Likewise, we will clearly note in the manuscript that this is important but considered for future work beyond this paper.
>
>   Here are the detailed experiments and results on ScanNet:
>
>   [*Adapt to monocular setting*] ScanNet contains 1,201 videos(scene) in the training set and 312 videos(scene) in the validation set. Adapting it for monocular 3D object detection, we utilized one RGB-D image per video, amounting to 1,201 RGB-D images for training and 312 for validation. We calculate the ground truth 3D bounding box label for each of our used views from their provided video(scene) level label (because some objects in the scene may not be visible in our monocular view).
>
>   [*Training and test*] For the baseline, we train an ImVoxelNet monocular 3D object detection model on the training set and test on the validation set. For our method, there are 8 overlap categories (sofa, bookshelf, chair, table, bed, desk, toilet, bathtub) in the 18 classes of ScanNet with our collected Objaverse data (main paper Table1). We use our 3D copy-past to augment the training set and train an ImVoxelNet. All the training parameters are the same as the training on SUN RGB-D dataset. We will release all the code.
>
>   Table: ScanNet experiments.
>
> |ScanNet AP_0.25| Average|bed |chair|sofa|table|bookshelf|desk|bathtub|toilet|
> |-|-|-|-|-|-|-|-|-|-|
> |ImVoxelNet|14.1|25.7|7.9|**13.2**|7.8|4.2|20.5|22.1|**11.5**|
> |ImVoxelNet+3D Copy-Paste|**16.9**|**27.7**|**12.7**|10.0|**10.8**|**9.2**|**26.2**|**29.2**|9.0|
>
> For the 2D recognition task, some related works [3,4] showed that simple 2D copy-paste is already good enough to help improve performance. While it is hard to conduct copy-paste in 3D, that is one of the motivations of our work. We believe our work should also help on 2D task and will treat it as future work.
>
> - **2D copy-paste baseline and individual object performance**:
>  For 2D insertion, it is hard to obtain the 3D bounding box, which is required for downstream monocular 3D object detection.
> The random insertion in the main paper comparison is 3D insertion, where the insertion position, size, pose, and illumination are not physically plausible, which causes a significant performance drop.
>
>   Here are the detailed SUN RGB-D monocular 3D object detection results of the main paper Table 2 on each individual object category:
>
> |SUN RGB-D AP_0.25 | Average (mAP)|bed |chair|sofa|table|book shelf|desk|bathtub|toilet|dresser|night stand|
> |--|--|---|---|--|--|--|--|--|--|--|--|
> | ImVoxelNet|40.96|72.0|55.6 |53.0|41.1|**7.6**|21.5|29.6|76.7|19.0|33.4|
> | ImVoxelNet+3D Copy-Paste|**43.79** |**72.6**|**57.1** |**55.1**|**41.8** |7.1|**24.1**|**40.2** |**80.7** |**22.3**|**36.9**|
>
> - **Include more datasets**:
>  Yes, we posit that a richer collection of objects to insert would be beneficial. However, we need full 3D models that we can insert in any pose (thus, the NYU dataset may not work as it does not provide full 3D object models). But other 3D object datasets (e.g., OmniObject3D) could be included in future work.
>
> - **Time cost to render one object**:
>  Overall it will take around 5~10 seconds. Specifically, searching the insertion position, pose, and size takes less than 0.5 seconds with iteration 1000. Plane detection, lighting estimation, and rendering take most of the time.
>
> - **Insert shiny or specular objects**:
> Changing the reflection property only influences the final rendering process; our physically plausible position, pose, size, and illumination are agnostic to the object's surface property. Specular objects will show reflections of other objects or lights during the rendering process.
>
> **References**
>
> [1] Dai, Angela, et al. "Scannet: Richly-annotated 3d reconstructions of indoor scenes." Proceedings of the IEEE conference on computer vision and pattern recognition. 2017.
>
> [2] Zhang, Cheng, et al. "Holistic 3d scene understanding from a single image with implicit representation." Proceedings of the IEEE/CVF Conference on Computer Vision and Pattern Recognition. 2021.
>
> [3] Dwibedi, Debidatta, Ishan Misra, and Martial Hebert. "Cut, paste and learn: Surprisingly easy synthesis for instance detection." Proceedings of the IEEE international conference on computer vision. 2017.
>
> [4] Ghiasi, Golnaz, et al. "Simple copy-paste is a strong data augmentation method for instance segmentation." Proceedings of the IEEE/CVF conference on computer vision and pattern recognition. 2021.

---

> > ### Comment · Reviewer_sM4b · 2023-08-16
> > **Response to author's rebuttal**
> >
> > Thank you for your comprehensive response addressing the concerns raised in my review. I appreciate the clarifications provided and the commitment to making the suggested changes in the revised version of the paper. I look forward to seeing the updated manuscript and the enhancements you plan to incorporate based on the feedback and other reviews.

---

> > > ### Author Response · Authors · 2023-08-21
> > > **Thank you for your response!**
> > >
> > > Thank you for your response! We will incorporate the suggested changes in the revised paper.

---

### Official Review · Reviewer_7hAU · 2023-07-06

**Soundness:** 3 good
**Presentation:** 2 fair
**Contribution:** 2 fair
**Rating:** 6
**Confidence:** 2

**Summary:**

The manuscript introduces system to create RGBD datasets augmented with additional 3d objects (one per rgb frame). The inserted objects are placed such that they dont intersect other objects, stand on the ground floor and the objects are rendered into the original RGB frame such that the lighting of the scene is respected. The augmented dataset can then be used to train slightly better 3d detectors since more 3d object examples can be supervised.

**Strengths:**

The manuscript tackles an important problem for 3D object detection research: the scarcity of supervision data and the difficulty inherent in augmenting in 3D. The 3 factors for physical plausability make sense to me and the implementation of them in the paper seems pretty solid judging by the example images from the generated augmented dataset.
The claim that the full system of physically plausible augmentation is needed is well supported by ablations against other plausible choices (random placement, simpler light sources). The system seems sound although I am not an expert on the most recent ways of estimating illumination.


**Weaknesses:**

The primary weakness to my mind is the use of only a single (small!) 3d object detection dataset with SUB RGB - ScanNet is substantially bigger and commonly used. Who knows maybe this method would be even more powerful on a harder dataset?

The delta in mAP is not very big with the addition of the presented augmentation method (from 40.96 to 43.79). I question whether people will want to implement the full presented system in order for such modest gains. This limits the potential impact in my eyes and means claim (2) "significant" improvements is not fully supported.

ImVoxelnet is not a single-view detector (it is multiview but can be used for single-view detection); In the original paper, the mAP for SUN RGB-D is actually 43.66 (I suppose thats because of the different IoU threshold of 0.15 but it raises the question to me why the evaluation in this manuscript doesnt use the same one?)

I am not 100% clear on the way the illumination is estimated and used as somebody who has never worked with such methods. This would make it hard for me to replicate the presented work.

There is a few sentences on semantic plausibility and that it didnt help but its not well explained how semantic plausibility was achieved. This is an important negative result that I am not sure I can trust at the current level of explanation.

**Questions:**

see weaknesses

**Limitations:**

Limitations such as the need for metrically scaled objects, the need for depth images and the knowledge of the gravity direction are not explicitly discussed in the paper.

---

> ### Author Rebuttal · Authors · 2023-08-09
>
> Thank you for your time and comments! Please see our response below.
>
> - **Experiments on ScanNet**:
> We use the SUN RGB-D dataset as this dataset offers 10,000+ monocular RGB-D images (scenes), and is densely annotated with 146,617 2D polygons and 58,657 3D bounding boxes. Many 3D object detection papers [1,2] use SUN RGB-D performance as the main result. ScanNet is a large-scale R-GBD video dataset that isn't specifically tailored for monocular 3D object detection. We appreciate the reviewer’s suggestion and have conducted new experiments on ScanNet.
>
>   Experimental settings are described in detail in the Global Response to all reviewers (above). We summarize it here:
>
>   [*Adapt to monocular setting*] We utilized one RGB-D image per video: 1,201 for training and 312 for validation. We calculate the ground truth 3D bounding box label for each of our used views from their provided video(scene) level label.
>
>   [*Training and test*] There are 8 overlap categories in ScanNet with our collected Objaverse data. We use our 3D Copy-Paste to augment the training set and train an ImVoxelNet. We show the average accuracy of the 8 overlap classes below. Our method improves ImVoxelNet by 2.8% mAP.
>
> |ScanNet AP_0.25| Average (mAP)|bed|chair|sofa|table|bookshelf|desk|bathtub|toilet|
> |-|-|-|-|-|-|-|-|-|-|
> |ImVoxelNet|14.1|25.7|7.9|**13.2**|7.8|4.2|20.5|22.1|**11.5**|
> |ImVoxelNet+3D Copy-Paste|**16.9**|**27.7**|**12.7**|10.0|**10.8**|**9.2**|**26.2**|**29.2**|9.0|
>
> - **Improvement significance and implementation easiness**:
> Monocular 3D object detection is a challenging task that requires inferring the 3D information given only a single RGB image, and improving from 40.96 to 43.79 can be considered significant as Reviewer DWmW also pointed out. If checking the performance leaderboard, it has been hard to improve mAP further beyond 40 (e.g., ImVoxelNet remains the current SOTA on Papers-With-Code on SUN RGB-D even though it is now 2 years old). Given our 2.83% improvement over that, our method is the new state-of-the-art performance on indoor scene monocular 3D object detection.
>
>   Beyond just a numerical improvement in mAP, our work provides a number of conceptual advances in the field, which we believe are important as well.  Firstly, the task of monocular 3D object detection is notably data-intensive, and labeling 3D labels can be both time-consuming and costly. Our approach addresses this challenge through data augmentation, introducing an automatic pipeline that remains model-agnostic. Since data augmentation is a one-off effort, it can potentially enhance various models. To allow easy usage, we will release our code, model, and generated data.
>   Secondly, through comprehensive experiments, we demonstrate the efficacy of our method across diverse models, such as ImVoxelNet and Implicit3D (detailed in the supplementary), and on different datasets, including SUN RGB-D and ScanNet. Importantly, our findings highlight the potential of 3D data augmentation in improving the performance of 3D perception tasks, opening up new avenues for research and practical applications.
>
> - **mAP 0.25 evaluation**:
> This has to do with the fact that official results are only available for a few combinations of dataset/object classes/IOU threshold: The official ImVoxelNet GitHub code, when using SUN RBG-D on “10 classes from VoteNet” setting (same as ours), also uses 0.25 IOU threshold with performance 40.7 (even though they use 0.15 with other datasets/classes).
> The authors provided an implementation in MMdetection3D Github [3], which also uses mAP 0.25 for the 10 classes from VoteNet setting (performance 40.96). We used the official code of MMdetection3D, so we used mAP 0.25. As you suggested, we also show our results with mAP 0.15 on SUN RGB-D dataset below. We will incorporate this in our revised paper.:
>
> |SUN RGB-D|mAP_0.15|
> |-|-|
> |ImVoxelNet|48.45|
> |ImVoxelNet + 3D Copy-Paste|**51.16**|
>
> - **Illumination estimation and reproducibility**:
> Illumination estimation is an important and challenging task, our main paper Sec. 2.3 (L108~L119) listed some representative works in this domain. We will release all our code and dataset upon acceptance for reproducibility.
>
> - **Semantic plausibility influence**:
> For this experiment, to make the insertion more globally semantically plausible (e.g., avoid inserting toilet into other rooms than bathroom), we only insert the object categories that already exist in the current room. For instance, if the room contains table and chair, we only consider inserting new objects that belong to these 2 categories.
>
>   The results (Table 4) show that considering the global semantic meaning (43.75) is on par with the random category selecting setting (43.79). One potential reason is that the downstream detection (CNN-based models) may rely more on local information to conduct detection, so they are not sensitive to the global semantics. Different from point cloud-based 3D detection, where context information is important as RGB information is often discarded, in monocular 3D object detection where the input is an RGB image, appearance per se may be the most important clue.
>
> - **Requirement of depth, scale, and gravity**:
> The metrically scaled object, depth, and gravity direction can be either dataset provided or estimated by off-the-shelf methods, e.g., metric depth estimation from ZoeDepth [4], and gravity direction estimation from ground segmentation with plane fitting.  We will add this discussion in the limitations section (currently, limitations are in the supplementary).
>
> **References**
>
> [1] Huang et al. "Perspectivenet: 3d object detection from a single rgb image via perspective points." NeurIPS 2019.
>
> [2] Zhang et al. "Holistic 3d scene understanding from a single image with implicit representation." CVPR 2021.
>
> [3] MMDetection3D: OpenMMLab next-generation platform for general 3D object detection.
>
> [4]"Zoedepth: Zero-shot transfer by combining relative and metric depth." arXiv 2023.

---

> > ### Comment · Reviewer_7hAU · 2023-08-14
> > **response**
> >
> > Thank you for addressing my questions and concerns. "To allow easy usage, we will release our code, model, and generated data." will be critical to enable others to benefit from this paper. Based on this promise and the clarification around the mAP improvement and the other reviewers responses I am happy to increase my rating to weak accept.

---

> > > ### Author Response · Authors · 2023-08-21
> > > **Thank you for your response!**
> > >
> > > Thank you for updating your review!

---

### Author Rebuttal · Authors · 2023-08-09

## Global response (common questions) to all reviewers

We would like to thank our reviewers, which put considerable time and thoughts for helping improve our paper.
We are pleased that the reviewers find our paper "very well-written and easy to follow"(R-sM4b, R-mHBD, R-DWmW); our method being described as "novel" (R-DWmW), "fully automated" (R-B4nT), "impressive, well-thought-of and designed" (R-sM4b) and "systematic and comprehensive"(R-mHBD), with a "solid implementation"(R-7hAU); our experiments were commended as "extensive" (R-DWmW), "solid" (R-B4nT), backed by "supportive ablations" (R-7hAU, R-mHBD), demonstrating "effectiveness" (R-sM4b), achieving "state-of-the-art performance" (R-sM4b, R-mHBD), and notably improving "monocular 3D detection by a large margin" (R-DWmW).

Please find below our summary of major changes and responses to some common questions. We will incorporate these major changes into our revised paper.

**Summary of major changes**:
1. [7hAU, DWmW, mHBD] We add new experiments on the ScanNet[1] dataset to show that 3D Copy-Paste also improves monocular 3D object detection performance on other dataset.
2. [7hAU, B4nT] We evaluate our monocular 3D object detection performance on SUN RGBD with mAP_0.15 and show consistent improvements.
3. [sM4b] We add detailed results of each individual object category on the SUN RGB-D dataset.


**Experiments on ScanNet dataset**:

To show the generalization to other datasets, we apply our 3D Copy-Paste to ScanNet[1] and conduct monocular 3D object detection with ImVoxelNet. ScanNet is a large-scale R-GBD video dataset that isn't specifically tailored for monocular 3D object detection. Here are the detailed experiments and results (we use ScanNet v2):

  [*Adapt to monocular setting*] ScanNet contains 1,201 videos (scenes) in the training set and 312 videos (scenes) in the validation set. Adapting it for monocular 3D object detection, we utilized one RGB-D image per video, amounting to 1,201 RGB-D images for training and 312 for validation. We calculate the ground truth 3D bounding box label for each of our used views from their provided video (scene) level label (because some objects in the scene may not be visible in our monocular view).

  [*Training and test*] For the baseline, we train an ImVoxelNet monocular 3D object detection model on the training set and test on the validation set. For our method, there are 8 overlap categories (sofa, bookshelf, chair, table, bed, desk, toilet, bathtub) in the 18 classes of ScanNet with our collected Objaverse data (main paper Table1). We use our 3D copy-past to augment the training set and train an ImVoxelNet. All the training parameters are the same as the training on SUN RGB-D dataset. We will release all the code.

  We show the results on the average accuracy of the 8 overlap classes (AP_0.25) in the Table below. Our 3D Copy-Paste improves ImVoxelNet by 2.8% mAP.

|ScanNet AP_0.25                   | Average (mAP)|bed |chair|sofa|table|bookshelf|desk|bathtub|toilet|
|-----------------|------|------|----|----|----|----|----|----|----|
| ImVoxelNet              |14.1 |25.7|7.9  |**13.2**|7.8  |4.2      |20.5|22.1   |**11.5**  |
| ImVoxelNet+3D Copy-Paste|**16.9** |**27.7**|**12.7** |10.0|**10.8** |**9.2**      |**26.2**|**29.2**   |9.0   |


**Experiments on other 3D detection method**:

To show the generalization of our method to other downstream monocular 3D object detection methods, in our supplementary material Table S1, we conducted additional experiments with another monocular 3D object detection model: Implicit3DUnderstanding (Im3D [2]). Using our method (Im3D + 3D Copy-Paste) improve the mAP_0.25 from 42.13 (Im3D) to 43.34.

**Performance details of each category in SUN RGB-D dataset**:

In the table below, we show the detailed SUN RGB-D monocular 3D object detection results with ImVoxelNet of the main paper Table 2 on each individual object category:

|SUN RGB-D AP_0.25                   | Average (mAP)   |bed |chair|sofa|table|book shelf|desk|bathtub|toilet|dresser|night stand|
|-----------------|------|------|----|----|----|----|----|----|----|----|----|
| ImVoxelNet              |40.96 |72.0|55.6 |53.0|41.1 |**7.6**      |21.5|29.6   |76.7  |19.0|33.4|
| ImVoxelNet+3D Copy-Paste|**43.79** |**72.6**|**57.1** |**55.1**|**41.8** |7.1      |**24.1**|**40.2**   |**80.7**  |**22.3**|**36.9**|


**Reference**


[1] Dai, A., Chang, A.X., Savva, M., Halber, M., Funkhouser, T. and Nießner, M., 2017. Scannet: Richly-annotated 3d reconstructions of indoor scenes. In Proceedings of the IEEE conference on computer vision and pattern recognition (pp. 5828-5839).

[2] Zhang, C., Cui, Z., Zhang, Y., Zeng, B., Pollefeys, M. and Liu, S., 2021. Holistic 3d scene understanding from a single image with implicit representation. In Proceedings of the IEEE/CVF Conference on Computer Vision and Pattern Recognition (pp. 8833-8842).

---

### Decision · Program_Chairs · 2023-09-21

**Decision:**

Accept (poster)

**Comment:**

All reviewers recommend acceptance. The AC sees no basis to overturn the reviews, and recommends acceptance. There was a discussion. Authors should attend to main points in the reviews, such as Reviewer 7hAU's questions about mAP (which was brought up by another reviewer during discussion), when preparing a final version.